# A Novel Methylene Blue Indicator-Based Aptasensor for Rapid Detection of *Pseudomonas aeruginosa*

**DOI:** 10.3390/ijms252111682

**Published:** 2024-10-30

**Authors:** Somayeh Maghsoomi, Julia Walochnik, Martin Brandl, Mai-Lan Pham

**Affiliations:** 1Center for Water and Environmental Sensors, Department for Integrated Sensor Systems, University for Continuing Education Krems, Dr.-Karl-Dorrek-Straße 30, 3500 Krems an der Donau, Austria; somayeh.maghsoomi.taramsari@donau-uni.ac.at (S.M.); martin.brandl@donau-uni.ac.at (M.B.); 2Institute of Specific Prophylaxis and Tropical Medicine, Medical University of Vienna, Kinderspitalgasse 15, 1090 Vienna, Austria; julia.walochnik@meduniwien.ac.at

**Keywords:** aptasensor, aptamer, biosensor, *Pseudomonas aeruginosa*, methylene blue

## Abstract

*Pseudomonas aeruginosa* is a significant opportunistic pathogen highly prevalent in the environment, requiring early detection methods to prevent infections in vulnerable individuals. The most specific aptamer for *P. aeruginosa*, F23, has been used for the development of various assays and sensors for early diagnosis and monitoring. In this study, a novel F23-based electrochemical aptasensor was designed using disposal gold screen-printed electrodes (Au-SPEs) with high reproducibility. Methylene blue (MB) was used as an exogenous indicator, which significantly amplified the electrochemical signal and improved the sensitivity of detection. The aptasensor explored a limit of detection (LOD) of 8 CFU·mL^−1^ and high selectivity for *P. aeruginosa* over other interfering bacteria. Furthermore, it showed potential to detect *P. aeruginosa* in tap water samples, offering a point-of-care tool for rapidly controlling the growth of this bacterium in various applications.

## 1. Introduction

*Pseudomonas aeruginosa*, a rod-shaped Gram-negative bacterium, is a clinically significant opportunistic pathogen listed by the World Health Organization as a priority pathogen. It can cause a wide range of severe acute and chronic infections, particularly in patients with preexisting medical conditions or weakened immune systems. Multidrug-resistant strains pose a particular threat in hospitals and other facilities with vulnerable individuals [1,2]. *P. aeruginosa* is ubiquitous in soil and aquatic environments, including man-made habitats, and can form biofilms on living and nonliving surfaces [3]. Rapid and accurate detection of *P. aeruginosa* is crucial for efficient monitoring. Numerous methodologies have been developed and applied for the detection and identification of *P. aeruginosa*, including microbiological cultivation methods [4], polymerase chain reaction (PCR)-based methods [5], immunological assays [4], and matrix-assisted laser desorption/ionization time-of-flight mass spectrometry (MALDI-TOP-MS) [6]. However, they are either time-consuming or require advanced equipment and specialized training [7].

In recent years, portable and easily operable biosensors have become valuable tools for various point-of-care applications in food safety, disease diagnostics, environmental monitoring, and detection of pathogens [8]. Among them, aptamer-based biosensors, or so-called aptasensors, have received particular attention due to their prolonged stability and cost-effectiveness [9]. This type of biosensor utilizes aptamers as alternative biorecognition elements instead of antibodies in conventional immunological biosensors. Aptamers are single-stranded DNA or RNA oligonucleotides developed in vitro through a process known as systematic evolution of ligands by exponential enrichment (SELEX) [10]. They have high affinity and specificity toward their targets ranging from small molecules to whole cells via their adaptive conformation changes [9,11]. So far, several specific aptamers for the detection of *P. aeruginosa* against the whole-cell bacterium [12,13,14] as well as its metabolites [15] have been identified. The highly specific aptamer F23, isolated by Wang and colleagues [12], has been widely used to develop various aptasensors for whole-cell detection of *P. aeruginosa* due to its low K_d_ value of 17.27 ± 5 nM. Fluorescent and electrochemical F23-based aptasensors are the most preferred platforms for these applications. For instance, a localized surface plasmon resonance (LSPR)-based sensing platform was developed employing F23 aptamer to identify whole-cell *P. aeruginosa* PAO1 strain with a limit of detection of 10 CFU·mL^−1^ [16]. Sarabaegi et al. introduced a highly sensitive electrochemical aptasensor for *P. aeruginosa* by covalently attaching the F23 to the surface of a glassy carbon electrode modified with nano-sized chitosan particles [17].

In the current study, we present a methylene blue-mediated aptasensor established from the F23 aptamer for the detection of *P. aeruginosa*. The F23 was utilized as a biorecognition probe of the target bacterium and attached to the surface of a disposal gold screen-printed electrodes (Au-SPEs) system. Methylene blue was supplemented to a working electrolyte as a redox indicator to enhance the detection sensitivity of the aptasensor.

## 2. Results

### 2.1. Characterization of the Aptasensor

The immobilization of the F23 aptamer on the surface of Au-SPEs was investigated by SEM and cyclic voltammetric analysis. Figure 1A shows the morphology of the surface of the bare SPEs observed by SEM, while Figure 1B shows the changes on the surface of the F23-modified SPEs. The modified electrode exhibited the tubular structures of the aptamer that were not observed on the surface of the bare electrode. These structures make the surface of the modified SPEs rougher than the bare one. Figure 1C shows the cyclic voltammograms of both, bare and modified SPEs, measured in Tris buffer solution (pH 6.0) with or without supplementation of 50 mM methylene blue (MB). Two redox peaks at around −0.2 V (anodic) and −0.35 V (cathodic) appeared in the voltammogram of the bare electrode in the presence of MB in the electrolyte, which are interpreted as redox peaks of MB (blue line, Figure 1C). These peaks were not observed on the bare electrode measured in the absence of MB in the electrolyte (black line, Figure 1C). The green and red lines in Figure 1C are the voltammograms of the F23-modified SPEs in the absence and presence of MB in the electrolyte, respectively. After F23 fabrication, the modified SPEs showed two oxidative (anodic) peaks at +0.11 V and +0.24 V determined in the electrolyte without the addition of MB (green line, Figure 1C). Nevertheless, a pair of redox peaks can be clearly seen at +0.11 V (anodic) and −0.21 V (cathodic) with high current in the presence of MB in the electrolyte (red line, Figure 1C).

Additionally, F23-modified SPEs showed good stability under storage conditions (at 4 °C in Tris buffer, pH 7.0) for up to 14 days with a loss of 13% of the signal compared with the first day. However, the electrochemical signal decreased to almost 50% after 21 days compared with the measurement on the first day of storage. After one month, the signal decreased to 38% (Appendix A).

### 2.2. Electrochemical Detection of P. aeruginosa

The analytical performance of the F23-modified SPEs was assessed by SWV using Tris buffer (pH 6.0) adding 50 mM MB in the presence of various concentrations of *P. aeruginosa* as described in the Materials and Methods section. Each measurement was performed in three different F23-modified SPEs. Figure 2A shows the electrochemical characteristic peak at around −0.21 V that appeared in the square wave voltammogram of modified SPEs incubated with different concentrations of the target bacterium. Additionally, obtained results point to a decrease in the peak currents observed with an increase in bacterial concentrations ranging from 0 to 10^10^ CFU·mL^−1^ because of a conformation change in the aptamer. A linear plot was established by the decrease in SWV peak currents obtained in the absence and presence of the target bacterium (ΔI) against the bacterial concentrations expressed as logarithm base 10 of CFU·mL^−1^. The resulting linear regression equation was determined as y = 10.754x + 156.13, with a correlation coefficient value (R^2^) of 0.9865 (Figure 2B).

The limit of detection (LOD) was calculated using the formula: LOD = 3∙(σ/m) where σ represents the standard deviation of the response and m is the slope of the obtained linear curve. The sensitivity was determined according to the formula: S = m/A with A corresponding to the area of the working electrode in SPEs. Based on the linear equation, the LOD and sensitivity of the created aptasensor were determined as 8 CFU·mL^−1^ and 85.349 μA Log [(CFU·mL^−1^)] ^−1^·cm^−2^ on an electrode’s surface area of 0.126 cm^2^, respectively.

### 2.3. Effect of Methylene Blue on Performance and Specificity of the Aptasensor

Table 1 presents the electrochemical signals measured from generated aptasensors incubated with several interfering bacteria at a concentration of 1.6 × 10^9^ CFU·mL^−1^, both in the absence or presence of the redox indicator in the electrolyte. The obtained electrochemical signals were expressed as the decrease in SWV peak currents (ΔI) obtained in the absence (as the control) and presence of the bacteria. The results revealed a significant enhancement in electrochemical signals when MB is utilized as a redox indicator in the working electrolyte. The measured signals of the aptasensors incubated with or without bacteria in Tris buffer (pH 6.0) containing MB increased by approximately 48% compared with those recorded in the electrolyte lacking MB. Furthermore, the created aptasensor demonstrated its specificity for *P. aeruginosa*, yielding a lower ΔI value compared with those obtained from other inferring bacteria. Holm–Sidak statistical analysis revealed a significant difference between the target bacterium and the other bacteria, with a *p*-value < 0.001 (Figure 3). On the other hand, the electrochemical signals obtained from all control bacteria were similar. There was no difference in the strength of the signals between these bacteria (one-way ANOVA: *n* = 12; F = 0.116, *p* = 0.917) (Figure 3). Notably, when incubating with a mixture of tested bacteria including *P. aeruginosa* and other interfering bacteria, the recorded signal was similar to that measured with *P. aeruginosa* only (Figure 3, *t*-test: *n* = 6, *p* = 0.430). This result indicated the selectivity of the generated aptasensor toward *P. aeruginosa* without any interference with other bacteria.

### 2.4. Evaluation of Tap Water Samples

Table 2 shows the application of the generated aptasensor for detection of *P. aeruginosa* in artificially contaminated tap water samples. The tap water was spiked with *P. aeruginosa* only and with a mixture of interfering bacteria. The concentrations of the bacteria in the samples were determined by the plate counting method and by the aptasensor. Compared with the plate counting method, the recovery rates from the measurements with the created aptasensor were 112% and 109% in the samples contaminated with *P. aeruginosa* only and with the bacterial mixture, respectively.

## 3. Discussion

In the present study, a novel aptasensor was established based on the most specific aptamer against *P. aeruginosa*, F23, which was developed by Wang et al. [12]. It is assumed that the presence of *P. aeruginosa* induces a conformation change in the stem-loop branch of the structure of F23 [12], causing an alteration of the electrode’s surface charge distribution. Our results are in line with this assumption as the current peak measured by SWV decreased following an increase in the bacterial concentration (Figure 2). Moreover, it was shown that supplementing MB to the electrolyte caused a significant enhancement in the SWV peak currents at around −0.21 V, compared with those obtained in the electrolyte lacking MB (Appendix A). This can be explained by the diffusion of MB on the gold surface of the SPEs and an electrochemical reduction via a heterogenous electron transfer from the electrode. MB is a well-known redox mediator for sensors and biosensors due to its electrostatic interaction with DNA and the binding of its planar aromatic core to minor and major groves of DNA, as well as its intercalation with the aptamer through π–π interaction [18]. For instance, MB has been used as a redox agent to strengthen the current signal for the highly sensitive detection of the α-synuclein oligomer as a biomarker for Parkinson’s and Alzheimer’s diseases [19] or to amplify signals of an electrochemical aptasensor for monitoring the dopamine concentration released from living Parkinson’s disease model cells [20]. Interestingly, shifts in potentials of redox peaks were observed in the cyclic voltammograms of both bare and the modified electrodes measured in the electrolyte containing or lacking MB (Figure 1). This suggests that both the F23 modification and the addition of MB affect the electrochemical characteristics of the electrode. Additionally, as an effort to overcome nonspecific adsorption, this aptamer was co-immobilized with MCH on disposable SPEs forming self-assembly monolayers (SAMs) on the gold surface owing to its alkanethiol group (Thiol-MC6) tagged at the 5′ end of the oligos [21]. The thiol-SAM has been utilized on gold materials as a building block for the establishment of sensors and biosensors based on the strength of the S–Au bond and the van der Waals interaction of the hydrocarbon chain that makes it highly stable [18]. Furthermore, the C6 linker at the 5′ end between the thiol group and the aptamer has been demonstrated to generate sufficient space for the optimal folding of the aptamer as well as binding to the target bacterium [18]. As a result, in the current study, the LOD of the created aptasensor was low with a value of 8 CFU·mL^−1^ and was comparable with those of other F23-based electrochemical aptasensors (Table 3). For instance, Das et al. introduced an aptamer-mediated colorimetric and electrochemical detection of *P. aeruginosa* in water based on the inhibition of the inherent peroxidase-like activity of gold nanoparticles (GNPs) in the presence of the target bacterium with an LOD of 60 CFU·mL^−1^ [22]. In another study, *P. aeruginosa* was detected with an aptamer/polyadenylated DNA interdigitated gold electrode piezoelectric sensor with an LOD of 9 CFU·mL^−1^ in buffer and 52 CFU·mL^−1^ in blood samples [23]. An aptamer-functionalized localized surface plasmon resonance sensor on a multisport gold-capped nanoparticle array ship exploited its LOD of 10^4^ CFU·mL^−1^ with *P. aeruginosa* [24]. However, the method of calculating the LOD varies slightly between the studies. In the current study, as in most other studies, it was determined based on the standard deviation of the response and the slope of the calibration curve. The generated aptasensor showed high specificity and selectivity for the target bacterium when tested with interfering bacterial species, including other *Pseudomonas* species, namely, *P. putida*. Notably, the aptasensor in this study was constructed using disposal Au-SPEs, which tend to give a low reproducibility with a relative standard deviation (RSD) greater than 5% in spite of its low cost and feasibility [25]. More importantly, consecutive measurements in the same SPEs resulted in a degeneration in the performance of the reference electrode [25]. Subsequently, each measurement in this study was carried out by three different aptasensors, with the RSD (relative standard deviation) value ranging from 0.01–2.5%. This low value points to high reproducibility of the studied aptasensor, compared with those of other SPEs-based sensors. For example, an aptasensor for freshwater cyanobacteria gave a lower reproducibility with an RSD value of 6.521% [26]. Another aptasensor for α-synuclein showed an RSD between 1.8 and 4.9% [19]. Regarding the feasibility of the presented aptasensor for monitoring *P. aeruginosa* in water samples, the recovery rates under both test conditions specified that this aptasensor is suitable to detect the target bacterium in water samples with high repeatability.

## 4. Materials and Methods

### 4.1. Bacterial Strains and Culture Conditions

*Pseudomonas aeruginosa* ATCC 27853 obtained from TCS Biosciences Ltd. (Buckingham, UK) was grown in the nutrient broth (NB) (CarlRoth, Vienna, Austria) at 37 °C, with 150 rpm shaking. *Escherichia coli* ATCC11330, *Pseudomonas putida* ATCC 49128, *Staphylococcus aureus* ATCC 12600, and *Enterococcus faecalis* ATCC 19433 from the culture collection of the University for Continuing Education Krems (UWK) were cultivated in the lysogeny broth (LB) (CarlRoth, Vienna, Austria) at 37 °C, with 150 rpm shaking.

### 4.2. Chemicals and Aptamer Sequence

6-mercapto-1-hexanol (MCH), methylene blue (MB), TCEP (tris-(2-carboxyethyl) phosphine hydrochloride), Tris (2-amino-2-(hydroxymethyl)-1,3-propanediol), Tris buffer solution, and bovine serum albumin (BSA) were purchased from Sigma-Aldrich (St. Louis, MO, USA). NaH_2_PO_4_, Na_2_HPO_4_, NaCl, and MgCl_2_ were purchased from CarlRoth (Vienna, Austria). A high-salt Tris buffer was used as a fabrication buffer containing 10 mM Tris buffer, 1 M NaCl, and 1 mM MgCl_2_ (pH 7.4) [29]. The TBS-T (Tris-buffered saline supplemented with 0.05% Tween 20) (Sigma-Aldrich, St. Louis, MO, USA) with adjusted pH 7.4 containing 1% BSA was employed as a binding buffer. A redox electrolyte was prepared with Tris buffer (pH 6.0) containing with or without 50 mM MB. Disposal gold screen-printed electrodes (Au-SPEs) (DS 220AT) were purchased from Metrohm Inula GmbH (Vienna, Austria). A specific alkanethiol-modified aptamer for *P. aeruginosa*, F23, with a length of 60 nucleotides and the following sequence: 5′-Thiol MC6-CCCCCGTTGCTTTCGCTTTTCCTTTCGCTTTTGTTCGTTTCGTCCCTGCTTCCTTTCTTG-3′ [12] was synthesized by IDT (Integrated DNA Technologies, Leuven, Belgium).

### 4.3. Fabrication of Electrochemical Screen-Printed Electrodes

The Au-SPEs were fabricated with thiolated F23 and prepared as previously described [26] with some modifications, as illustrated in Figure 4. Briefly, 1 µM of thiolated F23 was initially incubated in 50 µL of fabrication buffer containing 100 µM TCEP for the reduction of disulfide-bonded oligos. The reaction was performed in 2 h at ambient laboratory room temperature (23 ± 1 °C) avoiding light. The reduced F23 was heated at 95 °C for 5 min and cooled on ice for 5 min to stabilize its conformation. Subsequently, 10 µL of treated F23 were dropped on the surface of Au-SPEs and the immobilization was carried out in a hydrated chamber at room temperature in the dark for 13 h. The unbound aptamers were removed by washing the electrode with deionized water. The resulting electrode was backfilled with 2 mM of MCH for 2 h at room temperature in the dark. Finally, the functionalized electrode was washed with deionized water and used immediately for the detection of the target bacterium. Alternatively, fabricated SPEs were stored in Tris buffer at 4 °C to evaluate the stability of the SPEs.

### 4.4. Electrochemical Measurements

All electrochemical measurements were performed using a redox electrolyte (Tris buffer, pH 6.0) with or without adding 50 mM methylene blue as described above for cyclic voltammetry (CV) and square wave voltammetry (SWV) in a Gamry Reference 600+ potentiostat (Gamry instruments, Warminster, PA, USA) at room temperature (23 ± 1 °C). The CV analysis was conducted within a scanning range of +1.0 V to −0.6 V, a scan rate of 50 mV·s^−1^, and an amplitude of 40 mV. The SWV measurements were set up with a scanning range from −0.6 to 0 V, a frequency of 50 Hz, a step size of 4 mV, and an amplitude of 40 mV. Each run was performed at least in triplicates using different F23-modified SPEs.

### 4.5. Assessement of the Aptasensor Fabrication

The immobilization of the thiolated F23 aptamer was verified by scanning electron microscopy (SEM) using a FlexSEM 1000 instrument (Hitachi, Düsseldorf, Germany) at an acceleration voltage of 20 kV and cyclic voltammetric analysis as described above. The stability of the fabricated SPEs was evaluated in the storage conditions at 4 °C in Tris buffer, pH 7.0. Every week, the SWV signals of three stored modified SPEs were determined as described above and compared with the initial signal measured on the first day of storage.

### 4.6. Analytical Performance of the Aptasensor

The target bacterium, *P. aeruginosa* ATCC27853, was cultivated following the cultivation conditions described above until the optical density at 600 nm (OD_600_) reached its exponential phase. A dilution series with different concentrations (10^1^ to 10^9^ CFU·mL^−1^) was prepared from the obtained bacterial culture. Bacterial cells from each dilution were harvested by centrifugation (4000× *g*, 4 °C, 5 min) and resuspended in the binding buffer. Subsequently, 10 µL of cell suspension of each dilution was dropped onto different disposal fabricated SPEs and incubated at 25 °C for 1 h. The electrodes were then washed with 100 µL of binding buffer followed by deionized water to exclude unbound cells. The SWV measurements were carried out in the redox electrolyte containing MB (50 mM) following the protocol described above (Figure 4). The bacterial concentration was determined by plating selected dilutions on LB agar plates. Plates were incubated at 37 °C overnight and grown colonies were counted. A number of colonies were expressed as colony-forming units (CFU) per ml (CFU·mL^−1^).

The selectivity and specificity were assessed by determining the SWV signal of the aptasensor binding to *E. coli*, *S. aureus*, *P. putida*, and *E. faecalis* as control microorganisms compared with that obtained from the binding to the target bacterium. All bacteria were grown in their suitable conditions as mentioned above and harvested at a concentration of 1.6 × 10^9^ CFU·mL^−1^.

### 4.7. Tap Water Analysis

The aptasensor in this work was used to detect *P. aeruginosa* in tap water. Briefly, tap water samples were spiked with *P. aeruginosa* as well as other control bacteria at a concentration of 1.6 × 10^9^ CFU·mL^−1^. The electrochemical signal of the created aptasensor was measured before and after being treated with tap water samples following the procedures described above.

### 4.8. Statistical Analysis

All experimental procedures and measurements were carried out in triplicates on three different modified SPEs. Statistical analyses were conducted utilizing Sigma Plot version 12.5 software (Systat Software Inc., San Jose, CA, USA), and the results are expressed as the mean ± standard deviation (SD) when appropriate.

## 5. Conclusions

In summary, this study provides an aptasensor for the detection of *P. aeruginosa* based on the F23 aptamer and methylene blue as an exogenous indicator. The aptasensor was designed for disposable SPEs with high reproducibility, offering a point-of-care testing tool for rapid monitoring of the target bacterium in various applications. By employing MB as a redox indicator, the studied sensor displayed high sensitivity with a low LOD value of 8 CFU·mL^−1^. Furthermore, it was demonstrated to be highly selective toward *P. aeruginosa*.

## Figures and Tables

**Figure 1 ijms-25-11682-f001:**
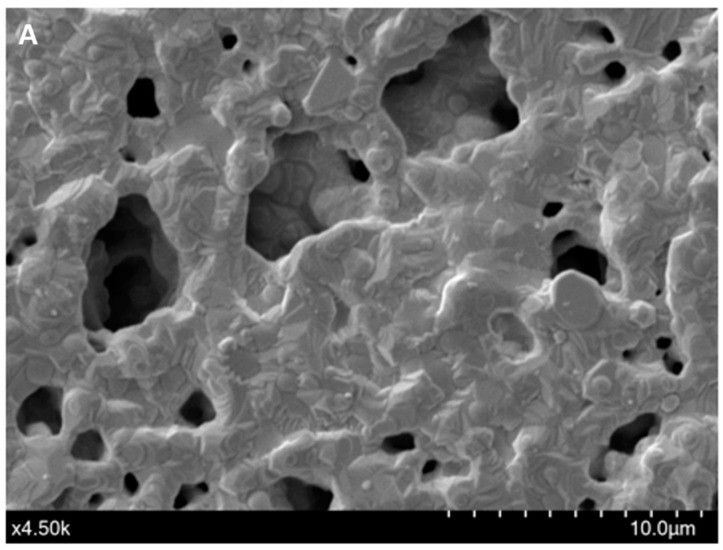
(**A**,**B**) SEM micrographs of the bare Au-SPEs (**A**); the modified Au-SPEs by the assembly of F23 aptamer on the surface of Au-SPEs (**B**). (**C**) cyclic voltammogram of the bare Au-SPEs (black) measured in the electrolyte lacking MB, the bare Au-SPEs measured in the electrolyte supplemented with MB (blue), the modified Au-SPEs measured in the electrolyte lacking MB (green), and the modified Au-SPEs measured in the electrolyte supplemented with MB (red) within scan range from +1.0 V to −0.6 V at a scan rate of 50 mV·s^−1^. The arrow indicates the scanning direction.

**Figure 2 ijms-25-11682-f002:**
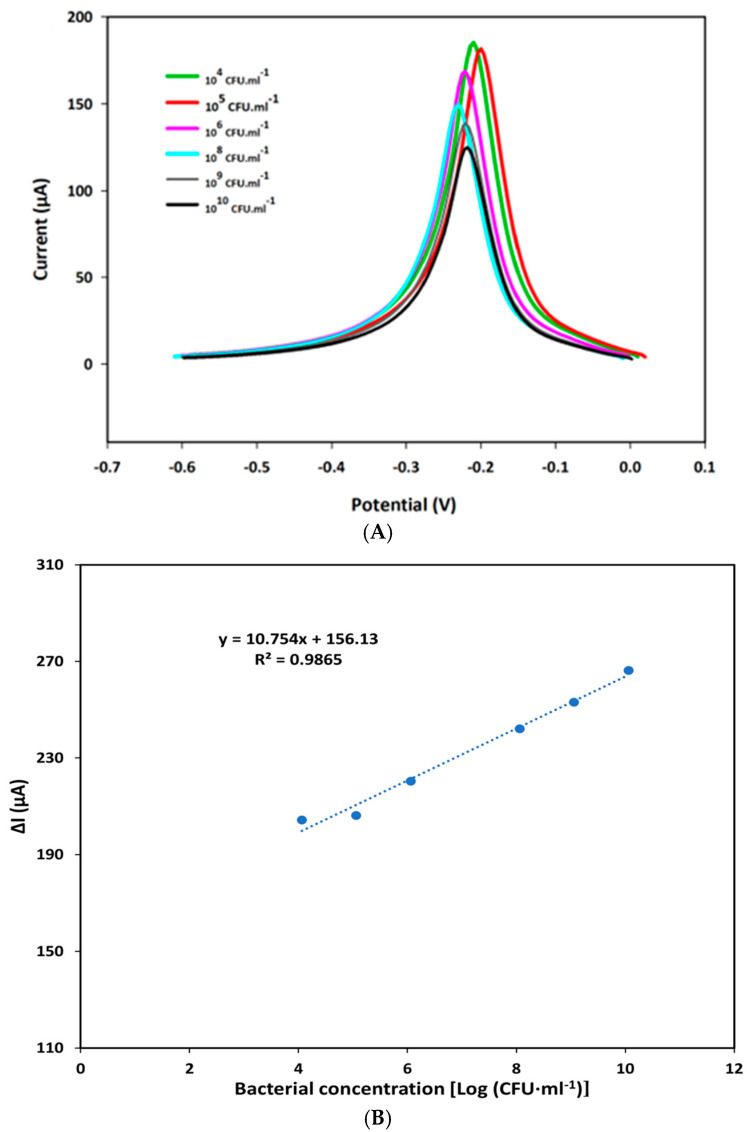
(**A**) SWV characteristic of the aptasensor in Tris buffer (pH 6.0) containing 50 mM MB after being incubated with various concentrations of *P. aeruginosa* ATCC27853 for one hour: (green) 10^4^ CFU·mL^−1^; (red) 10^6^ CFU·mL^−1^; (pink) 10^7^ CFU·mL^−1^; (blue) 10^8^ CFU·mL^−1^; (gray) 10^9^ CFU·mL^−1^; and (black) 10^10^ CFU·mL^−1^. (**B**) Linear calibration curve of the aptasensor plotted by the decrease in peak current (ΔI) measured by SWV against the concentration of bacterium expressed in logarithm base 10 of CFU per ml (CFU·mL^−1^). The ΔI was calculated by subtraction of the measured current in the absence of the target bacterium and the current determined in the presence of the target bacterium.

**Figure 3 ijms-25-11682-f003:**
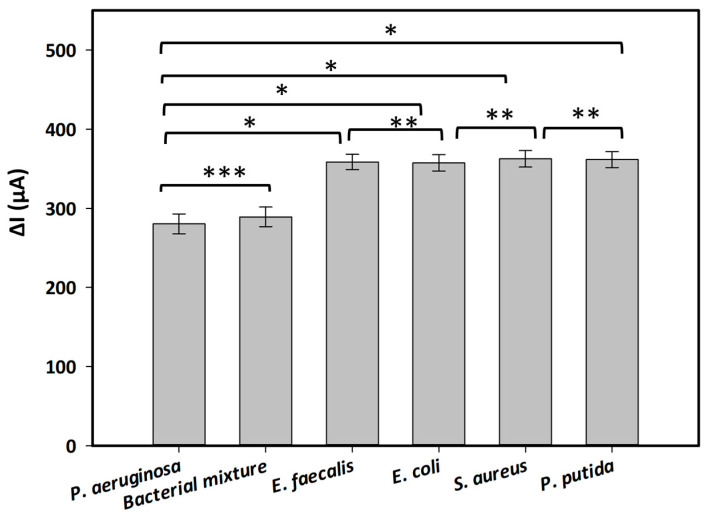
Selectivity and specificity of the aptasensor in the presence of interfering bacteria, including *P. aeuruginosa*, *E. coli*, *E. faecalis*, *S. aureus*, *P. putida*, and a mixture of tested bacterial strains. Each bacterium was harvested at a concentration of 1.6 × 10^9^ CFU·mL^−1^. Statistical analyses were performed with: (*) *p* < 0.001; (**) *p* = 0.917; (***) *p* = 0.430.

**Figure 4 ijms-25-11682-f004:**
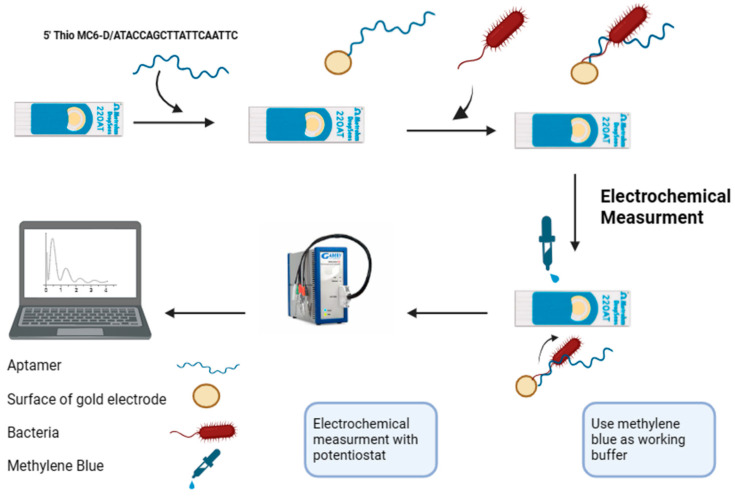
A scheme illustrating the fabrication and electrochemical detection process of the aptasensor using disposal screen-printed gold electrodes (Au-SPEs) (created by https://www.biorender.com/, accessed on 17 January 2024). The treated aptamer F23 was immobilized onto the gold surface of the Au-SPEs via gold-thiol SAM. Subsequently, the fabricated Au-SPEs were incubated with the target bacterium, *P. aeruginosa*, at 25 °C for one hour. Electrochemical measurements were then performed in an electrolyte (Tris buffer, pH 6.0) containing 50 mM methylene blue.

**Table 1 ijms-25-11682-t001:** Electrochemical signals obtained from aptasensors incubated with interfering bacteria and measured in the electrolyte supplemented with or without MB.

	ΔI (µA) *^a^*		Signal Gain (%) *^b^*
	Without MB	With MB	
** *P. aeruginosa* **	133.931 ± 5.155	280.387 ± 12.633	47.633 ± 2.175
**Bacterial mixture *^c^***	134.272 ± 4.714	283.433 ± 9.776	47.346 ± 1.651
** *E. coli* **	173.531 ± 5.879	358.644 ± 9.699	48.389 ± 1.347
** *E. faecalis* **	172.001 ± 6.148	357.479 ± 10.363	48.121 ± 1.349
** *S. aureus* **	178.024 ± 5.606	362.551 ± 10.466	49.112 ± 1.309
** *P. putida* **	176.440 ± 5.66	361.563 ± 10.066	48.807 ± 1.314

***^a^*** ΔI is defined as the decrease in peak currents obtained in the absence (control) and presence of bacteria as measured by SWV. The F23-modified SPEs, incubated with or without interfering bacteria, were electrochemically analyzed in electrolytes containing MB and lacking MB. The measured peak currents of treated aptasensors are given in Appendix A. ***^b^*** Signal gain is the percentage of ΔI value calculated from measurements in the electrolyte containing MB to those in the electrolyte lacking MB. ***^c^*** The aptasensor was incubated with a mixture of tested interfering bacterial strains, including *P. aeruginosa*.

**Table 2 ijms-25-11682-t002:** Analytical performance of the aptasensor in artificially contaminated water.

Contaminated Tap Water Samples	Bacterial Concentration by Plate Counting [Log (CFU·mL^−1^)]	Measured Concentration by Aptasensor [Log (CFU·mL^−1^)] **	Recovery Rate (%) *
With *P. aeruginosa* only	9.204	10.305 ± 0.147	112 ± 1.602
With bacterial mixture	9.204	10.017 ± 0.159	109 ± 1.737

* The recovery rate is the percentage of bacterial concentration expressed in Log (CFU·mL^−1^) measured by aptasensors to the initial bacterial concentration added to the water samples, which was determined by the plate counting method. ** The bacterial concentration measured by the created aptasensor is calculated based on the created linear equation (Figure 2B).

**Table 3 ijms-25-11682-t003:** Comparison of the analytical performance of various F23-based electrochemical aptasensors for *P. aeruginosa*.

Method	Material	LOD (CFU·mL^–1^)	References
LSPR	Nanosphere lithography	10	[16]
LSPR	MG-NPA	104	[24]
Aptamer–nanozyme–electrochemical	Gold nanoparticles	60	[22]
Piezoelectric/magnetoresistive	Au IDE-MSPQC	52 (in blood samples) 9 (in buffer)	[23]
Electrochemical	AgNPs/GCE	33	[27]
Electrochemical	CSPEs/MIL-101 (Cr)/MWCNT	1	[28]
Electrochemical	Au-SPEs	8	In this study

## Data Availability

Data is contained within the article and Appendix A.

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
