# Peer review of "A Novel Methylene Blue Indicator-Based Aptasensor for Rapid Detection of Pseudomonas aeruginosa"

_ijms, 2024, doi:10.3390/ijms252111682_

Round 1

Reviewer 1 Report

Comments and Suggestions for Authors

In this paper, authors reported an electrochemical sensor based on aptamer for detecting Pseudomonas aeruginosa in tap water. The sensor was based on the biochemical structure and specificity of Pseudomonas aeruginosa, and an Au-SPE-F23 aptamer based electrochemical immunoassay sensor was designed. The detection signal of the sensor was enhanced by using methylene blue as an exogenous indicator. The idea is interesting and lots of work had been done for the verification of the sensing mechanism. However, there are still some questions need to be clarified.

Here are some comments for your references.

#1. The sizes of the illustrations in the article are inconsistent. Please check the whole manuscript carefully.

#2. Figure 1 was labeled as SEM micrographs of (a), (b), (c), which can be understood as the three figures in Figure 1 being SEM micrographs, but it is clear that figure (c) is a CV graph.  There should be clear annotations on the figure that include SEM micrographs and CV graph. Furthermore, the labeling of (a), (b), (c) in the caption should be matched the labels in the figure (A), (B), (C).

#3. The SEM micrographs only show the changes of electrode surface morphology before and after modification, but does not point out the specific position and structure of F23 aptamer in the SEM micrograph.

#4. Can aptamers be observed at a magnification of 1000x in a scanning electron microscope?

#5. An obvious problem is this paper lacks the repeatability experiment of the electrode, so it cannot prove that the results shown by the detection of the electrode are not accidental.

#6. This article does not explain why the peak voltage positions of the target bacteria in Figure 2A were different, and whether this phenomenon has any effect on the detection results.

#7. The value of "error bar" is not marked in the results of Figure 2B, so the goodness of fit curve lacks of reliability. It is suggested to conduct multiple sets of experiments and calculate "error bar".

#8. The horizontal and vertical coordinates of Figure 2B lack of scale marks.

#9. Figure 3 unspecified unit of current.

#10. In the selectivity experiment, the currents in the detection results of the target bacteria and other bacteria were different. But it could be seen from the results that the sensor had current response to other bacteria, proving that the sensor could also detect other bacteria. In the detection of unknown samples, it is impossible to prove whether the species of bacteria present in the sample is the target bacteria.

#11. The discussion section is advised to be divided into distinct segments, with each focusing on a specific theme or point of view, to enhance the clarity and logic of the article.

#12. In section 4.4, the author explained that all electrochemical measurements were performed using a redox electrolyte containing methylene blue. However, in the discussion in section 2.1, a redox electrolyte without methylene blue was used to detect the electrochemical performance of the electrode. Inconsistency between the preceding and following sentences.

#13. The article does not explain how to determine the optimal dosage of methylene blue solution, and there is no experimental process and data support.

Author Response

Thank you for spending your time to review our manuscript. Please find the detailed responses below and the corresponding revisions that were highlighted in the re-submitted file.

#1. The sizes of the illustrations in the article are inconsistent. Please check the whole manuscript carefully.

Answer: we improve the illustrations.

#2. Figure 1 was labeled as SEM micrographs of (a), (b), (c), which can be understood as the three figures in Figure 1 being SEM micrographs, but it is clear that figure (c) is a CV graph.  There should be clear annotations on the figure that include SEM micrographs and CV graph. Furthermore, the labeling of (a), (b), (c) in the caption should be matched the labels in the figure (A), (B), (C).

Answer: We improve the figure caption.

#3. The SEM micrographs only show the changes of electrode surface morphology before and after modification, but does not point out the specific position and structure of F23 aptamer in the SEM micrograph.

Answer: The SEM can only used to observe the modification on the electrode’s surface before and after the immobilization of the aptamer. The observation of the aptamer’s structure and its immobilization on the surface of electrode is usually carried out by AFM (atomic force microscopy). Due to the limitation of our laboratory, we can only use SEM to observe the modification of the electrode’s surface. As you can see in Figure 1A and B, the electrode’s surface after the immobilization of aptamer became rougher than the bare electrode. On the surface of modified electrode, tubular structures appeared. They are believed as immobilized aptamers. This result was described in the results section.

#4. Can aptamers be observed at a magnification of 1000x in a scanning electron microscope?

 Answer: The magnification used to observe the bare and modified electrodes is 4500x. The magnification of 1000x is low to observe the modification on the surface of the electrode.

#5. An obvious problem is this paper lacks the repeatability experiment of the electrode, so it cannot prove that the results shown by the detection of the electrode are not accidental.

Answer: In this work, we used disposal screen-printed electrodes (SPEs) which was intentional to ensure that each experiment was conducted on a fresh, uncontaminated surface. The SPEs are single-use, allows precise and reproducible conditions. The continuous application of potential in the same SPEs can make the degeneration in the performance of the reference electrode. Therefore, in this study as well as in other study related to SPEs, the electrochemical measurements were performed in different SPEs to ensure that the outcomes are not influenced by any residual materials or degradation of RE, making the results more reliable and accurate. These are some literatures about the SPEs: ‘G. Paimard, E. Ghasali, and M. Baeza, "Screen-printed electrodes: Fabrication, modification, and biosensing applications," Chemosensors, vol. 11, no. 2, p. 113, 2023, doi: 10.3390/chemosensors11020113’. ‘Taghdisi, S.M.; Danesh, N.M.; Nameghi, M.A.; Ramezani, M.; Alibolandi, M.; Hassanzadeh-Khayat, M.; Emrani, A.S.; Abnous, K. A novel electrochemical aptasensor based on nontarget-induced high accumulation of methylene blue on the surface of electrode for sensing of α-synuclein oligomer. Biosens. Bioelectron. 2019, 123, 14–18, doi:10.1016/j.bios.2018.09.081’.

The manufacturer of SPEs also instructed customers to single-use their disposal SPEs. We explained and mentioned regarding to the measurement in three different SPEs in the discussion section. We also discuss the reproducibility of the modified SPEs in this section.

#6. This article does not explain why the peak voltage positions of the target bacteria in Figure 2A were different, and whether this phenomenon has any effect on the detection results.

Answer: In Figure 2A, the variation in peak voltage positions for the target bacterium can be attributed to slight differences in the electrochemical response due to several factors such as bacterial concentration, the electrode’s materials, immobilized buffer, pH of buffer, the working electrolyte, temperature conditions, and interaction dynamics between the aptasensor and the target bacterium. The obtained current peak became smaller and wider, resulting in a shift in their potential. And these shifts are typical in biosensors studies and do not adversely affect the detection results. The detection method relies on the magnitude of the current response rather than the exact peak position because the interaction between the aptamer and the target bacterium generates a detectable signal within a certain voltage range. In this work, the shifts in peak voltage remain within an acceptable range, and they do not compromise the reliability or sensitivity of the detection process. We verified this consistency through multiple replicated (at least in triplicate).

#7. The value of "error bar" is not marked in the results of Figure 2B, so the goodness of fit curve lacks of reliability. It is suggested to conduct multiple sets of experiments and calculate "error bar".

Answer: Figure 2B presented the linear calibration curve of the aptasensor. All measurement were conducted at least in triplicate and the results were expressed as the mean of at least three measurements. Both the standard deviation and error were calculated for each point. However, to maintain visual clarity and avoid clutter in Figure 2B, we removed the error bars. We assure that the error values were minimal, and their removal does not affect the reliability of the calibration curve.

#8. The horizontal and vertical coordinates of Figure 2B lack of scale marks.

Answer: We improved the figure and added the scale marks.

#9. Figure 3 unspecified unit of current.

Answer: we improve the figure and added the unit of the current (µA).

#10. In the selectivity experiment, the currents in the detection results of the target bacteria and other bacteria were different. But it could be seen from the results that the sensor had current response to other bacteria, proving that the sensor could also detect other bacteria. In the detection of unknown samples, it is impossible to prove whether the species of bacteria present in the sample is the target bacteria.

Answer: The electrochemical signals detected in the presence of other control bacteria were due to the redox activity of methylene blue (MB), which is a well-known redox medicator. Therefore, in the samples that contained other control bacteria or in the absence of the bacteria, the current measured corresponds to the baseline electrochemical response of methylene blue. This does not indicate that the sensor can detect other bacteria. The detection of the target bacterium, Pseudomonas aeruginosa, is based on the specific interaction with the aptamer F23, which induces a conformational change when the target is present. This conformational change decreases the electrode’s surface charge distribution, leading to a reduced current response. This selective mechanism ensures that only the target bacterium causes a significant reduction in current, which is the basis for aptasensor’s specificity, as explained in the discussion section. Therefore, the current responses observed with other bacteria do not interfere with the accurate detection of the target bacterium.

#11. The discussion section is advised to be divided into distinct segments, with each focusing on a specific theme or point of view, to enhance the clarity and logic of the article.

Answer: In this manuscript, the results and discussion are presented separately. The results were already divided into distinct subparts, with detailed descriptions of the findings. Since the discussion flows logically, summarizing and addressing key findings directly to the results, we believe it is unnecessary to further divide the discussion into additional subparts. This structure maintain clarity while allowing for a cohesive interpretation of the results. Nonetheless, we appreciate the suggestion and will further improve it enhancing readability.

#12. In section 4.4, the author explained that all electrochemical measurements were performed using a redox electrolyte containing methylene blue. However, in the discussion in section 2.1, a redox electrolyte without methylene blue was used to detect the electrochemical performance of the electrode. Inconsistency between the preceding and following sentences.

Answer: The key method for electrochemical measurement to detect the target bacterium in our work utilizes an electrolyte containing methylene blue. Therefore, in the methods section, we specifically describe the electrochemical measurements performed with the electrolyte containing methylene blue. However, in the subpart 2.1 of the results section, we used an electrolyte without methylene blue as a control experiment to compare and assess the effect of the methylene blue on the measurement. This comparison was done to highlight the critical role of the methylene blue in enhancing the detection performance of the aptasensor. We appreciate your observation and hope this clarification resolves any perceived inconsistency. We revised information of the measurement in an electrolyte without methylene blue in the revised version to clarify your comment.

#13. The article does not explain how to determine the optimal dosage of methylene blue solution, and there is no experimental process and data support.

Answer: The concentration of methylene blue commonly used in electrochemical applications is 10-100 mM. We conducted preliminary tests using various concentrations of methylene blue, referencing multiple literatures to optimize the used concentration. Based on our results, we determined that a concentration of 50 mM of methylene blue provided the best signal and was selected for our investigations in this manuscript.

Reviewer 2 Report

Comments and Suggestions for Authors

This manuscript describes the preparation and performance of an novel electrochemical sensor for detection of Pseudomonas aeruginosa. The authors prepared F23 aptamer on Au electrode and used it in conjunct with MB as electrolyte. Signal enhancenment was observed in the presence of bateria. The LOD and sensitity of the sensor was calculated. The perfomance of the sensor system was also evalutated with complex systems like tap water samples. This is a novel method invention and I would recommend it for publication after addressing the following comments:

1. There are two cathodic peaks in the CV of bare Au-SPE lacking MB (Figure 1C, black trace). Unmodified electrode should be inert at this mild potential range. Could the authors elaborate on the origin of these peaks?

2. The characteristic voltammetry signal used in the detection is the -0.21 V cathodic peak for the F23-Au-SPE in the presence of MB. Without MB, the F23-Au-SPE does not give this characteristic peak in CV. Could the authors elaborate on the signal enhancement by bateria in the absence of MB? What's the electrochemcial character of the peak being detected? 

3. Following on the previous comment, for the SWV without MB, is there a potential with more prominent F23-Au-SPE electrochemcial signal for detection of bateria than the -0.21 V used as comparison in the study?

4. It's not clear to me whether the sensor give higher baterial concentration results from the artificial com tap water sample is coming from a more accurate measurement or error. Could the authors provide the real bateria concentration of this artifical contaminated sample?

Author Response

Thank you for spending your time to review our manuscript. Please find the detailed responses below.

  1. There are two cathodic peaks in the CV of bare Au-SPE lacking MB (Figure 1C, black trace). Unmodified electrode should be inert at this mild potential range. Could the authors elaborate on the origin of these peaks?

Answer: While the unmodified electrode is expected to be relatively inert in this mild potential range, the appearance of two cathodic peaks in the CV of the bare Au-SPEs lacking MB could originate from the reduction of adsorbed species or other electroactive compounds present on the gold surface. Additionally, the physical characteristics of the electrode, including surface roughness and morphology could also contribute to unexpected peaks as these features can influence charge transfer and mass transport phenomena. In this study, we directly used disposal Au-SPEs without cleaning step. As recommendation from the manufacture, the SPEs do not need to clean before use. We also performed a preliminary test to clean the SPEs with 0.5 M H2SO4, but there is no difference in the obtained redox behavior between uncleaned and cleaned SPEs.

  1. The characteristic voltammetry signal used in the detection is the -0.21 V cathodic peak for the F23-Au-SPE in the presence of MB. Without MB, the F23-Au-SPE does not give this characteristic peak in CV. Could the authors elaborate on the signal enhancement by bateria in the absence of MB? What's the electrochemcial character of the peak being detected? 

Answer: The characteristic -0.21 V peak observed for the modified SPEs in the presence of MB indicates the redox activity facilitated by this mediator. In the absence of MB, the modified SPEs does not exhibit this specific peak, which suggests that electrochemical response is closely tied to the presence of MB. Without MB, the electrochemical behaviour of the modified SPEs primarily relies on the direct interactions between the immobilized aptamer on the electrode and the target bacterium. However, without a redox mediator, the electrochemical response was significantly reduced, making it challenging to detect the bacterium through standard voltametric methods. We presented table 1 for comparing the signals obtained from aptasensors incubated with bacteria in the absence and presence of MB in the electrolyte. The signal can gain approx. 48% in the measurement when MB is presented in the electrolyte. The detected cathodic peak represents a cathodic reaction associated with the reduction of methylene blue as it accepts electron. This reduction is influenced by the binding events between the aptamer and the target bacterium, leading to changes in the electron transfer kinetics and resulting the observed peak in the CV. As described in the manuscript, in the electrolyte containing MB, the peak was detected at -0.21 V by SWV measured in applied scanning range. Without MB in the electrolyte, we observed much smaller peak at approximately -0.13 V by SWV, indicating a much weaker electrochemical signal. It is noted that the

  1. Following on the previous comment, for the SWV without MB, is there a potential with more prominent F23-Au-SPE electrochemcial signal for detection of bateria than the -0.21 V used as comparison in the study?

Answer: As mentioned in previous answer, in the absence of MB, the electrochemical response of the modified SPEs is significantly diminished, as indicated by the smaller peak observed at around -0.13 V by SWV in a scanning range of -0.6 to 0V, a frequency of 50 Hz, a step size of 4 mV and an amplitude of 40 mV. It is suggested that the direct detection of the bacterium without MB is inherently limited in sensitivity. Under the set-up parameters for SWV measurement as mentioned above, there is only one distinct peak at -0.21 V that was observed. This peak is associated with the reduction of MB, which facilitates the detection of the target bacteria through enhanced electrochemical signals. In the absence of MB, the smaller peak at -0.13V likely represents the signal from the direct interaction between the immobilized aptamer and the target bacterium. However, this signal is significantly weaker and may not be as reliable compared to the response obtained with MB. The current peaks measurement by SWV in the Tris buffer as the electrolyte in the absence and presence of MB were presented in the Table S1 in the supplementary data.

  1. It's not clear to me whether the sensor give higher baterial concentration results from the artificial com tap water sample is coming from a more accurate measurement or error. Could the authors provide the real bateria concentration of this artifical contaminated sample?

Answer: In the artificial contaminated tap water, we prepared the sample by spiking it with a known amount of the target bacterium (in the sample with only P. aeruginosa) and known amounts of bacteria (in the sample with bacterial mixture). All bacteria were inoculated to the tap water at the same concentration as 1.6 x 109 CFU·ml-1 (the concentration of each bacterium in the tap water is 1.6 x 109 CFU·ml-1). This value was converted to Log (CFU·ml-1), which is equivalent to a value of 9.204 as described in the manuscript. This known concentration was determined by plate counting and was used as a reference to evaluate the sensor’s accuracy. The detected concentration by the aptasensor was determined using SWV measurement and was calculated based on the equation obtained from the calibration curve (Figure 2B) of the aptasensor. The concentration was expressed as Log (CFU·ml1). The detected concentration might slightly differ due to the inherent variability in electrochemical measurements and the high sensitivity of the aptasensor toward the target bacterium, but overall, the results were consistent with expected outcomes based on the sensor’s calibration curve. The recovery rate of 112% for sample containing only target bacterium and 109% for the mixed bacteria sample remain within a reasonable range and suggest that the aptasensor provides reliable detection with a slightly positive bias.

Round 2

Reviewer 1 Report

Comments and Suggestions for Authors

I have looked through the revised manuscript and thought the mentioned comments have all been improved. So, I suggest acceptance of this paper for the publication in IJMS.